# A value chain analysis of interventions to control production diseases in the intensive pig production sector

Jarkko Niemi[1], Richard Bennett[2], Beth Clark[3], Lynn Frewer[3], Philip Jones[2], Thomas Rimmler[1], Richard Tranter[2]*

**1** Natural Resources Institute Finland (Luke), Seinäjoki, Finland, **2** School of Agriculture, Policy and Development, University of Reading, Reading, England, United Kingdom, **3** School of Natural and Environmental Sciences, Newcastle University, Newcastle upon Tyne, England, United Kingdom

* r.b.tranter@reading.ac.uk

**Data Availability Statement:** All relevant data are within the paper and its Supporting Information files.

## Abstract

Value chain analysis (VCA) calculated the financial effects on food chain actors of interventions to improve animal health and welfare in the intensive pig sector. Two interventions to reduce production diseases were studied. A generic chain diagram of linkages between stakeholders and value-added dimensions was designed. Data on structure and financial performance were collected for the sector. The production parameters and financial effects of the interventions were then described to illustrate impact on the supply chain. The effects of the interventions were also assessed at market level using economic welfare analysis. The sectors in Finland and the UK are small in farm numbers and few companies produced much of the output in a largely vertically-integrated structure. The most beneficial intervention in financial terms to farmers was improved hygiene in pig fattening (around +50% in gross margin). It was calculated to reduce the consumer price for pig meat by up to 5% when applied at large, whereas for improved management measures, it would reduce consumer price by less than 0.5%. However, the latter added value also through food quality attributes. We show that good hygiene and animal care can add value. However, evaluation of the financial and social viability of the interventions is needed to decide what interventions are adopted. The structure of supply chains influences which policy measures could be applied. Of the two interventions, improved pig hygiene had the largest potential to improve efficiency and reduce costs. The studied interventions can also provide new business opportunities to farms, slaughterhouses and food sector companies. More evidence is needed to support public policies and business decision-making in the sector. For this, evidence on consumer attitudes to production diseases is needed. Nevertheless, the study makes an important contribution by showing how improvements in health and welfare benefit the whole chain.

**Funding:** This work was conducted under the PROHEALTH project which received funding from the European Union's 7th Framework Programme for Research, Technological Development and Demonstration under Grant Agreement Number 613574 to JN. The funder had no role in study design, data collection and analysis, decision to publish or preparation of the manuscript.

**Competing interests:** The authors have declared that no competing interests exist.

## Introduction

Pig supply chains are an important part of the food sector. Between 2014–2017, the EU pig sector produced 23.1B kg of pig meat on average per year [1] and the value of annual pig meat production in the EU at the farm gate price was €35.5B. However, the sector is faced by the risks of production diseases. Production diseases originate from a complex interaction between the pathogen (where present), the animal and the environment where it is kept. These diseases can negatively affect production and animal welfare and they can lead to substantial financial losses for pig producers [2,3]. Research shows that management and environmental factors influence these diseases [4,5]. Hence, producers seek interventions which can be used to control production diseases.

To implement these interventions, it is important to understand how stakeholders view them and how they could influence their businesses. Addressing socio-economic aspects related to the interventions can help to understand the rationale of adopting animal health management practices. Livestock producers play an important role in this, but focusing only on farmers is too narrow an approach because the acceptance of production practices, including disease management, among consumers and other stakeholders is essential for the sustainability of livestock production.

A value chain is a set of inter-connected activities, individuals or businesses that transport and transform a raw material from the original producer to the final consumer [6]. Value Chain Analysis (VCA) analyses the nature and sources of added value within a supply chain, and the potential for reducing inefficient use of resources in it, with the focus explicitly on the determinants of value within a manufacturing process rather than simply measuring the outputs of the process [7].

The aim of the study reported here was to analyse value effects of specific disease mitigation scenarios in the pig production chain. Two sets of interventions targeted on different challenges on pigs were defined for use in the VCA using results of experiments on farms studied in the PROHEALTH project. The two intervention scenarios for pigs were: improved hygiene in pig fattening, and better care of piglets and sows to improve piglet viability. The interventions were analysed to identify how supply-side experts and citizens viewed the interventions, and to illustrate their financial significance to stakeholders. The purpose of this analysis was to identify possible barriers to the adoption of specific interventions, as well as demand pull, and supply push, factors which would favour adoption of the interventions. Possible effects of the studied interventions were examined in both Finland and the UK. The results will enable quantification of the extent to which society and the animal production sector benefits from the interventions, and whether there are new business opportunities to promote competitiveness, resilience and sustainability. This study is novel for, as little past work has focused on the financial effects of pig production diseases, or the views of consumers regarding controlling them. As such, the study will make an important contribution to the pig sector by showing how improvements in health and welfare of pigs might benefit the whole value chain. This will be of great relevance to policy makers.

In the section which follows, we outline: the interventions examined; sources of information used in the analysis; and methods employed in the analysis. The Results section characterises value-creation potential from the view of financial impacts, sector-level impacts and consumer views on the interventions and, by discussing the business opportunities that could arise, based on the characteristics of the interventions themselves, what needs to be taken into account. Then we discuss the results of the studied interventions and, finally, draw conclusions from our research.

## Materials and methods

### Configuring the value chain

VCA can be carried out at different levels including: individual product or commodity; firm; sector; and at various geographical levels e.g. region, country or global [8].

According to [9] and [10], VCA has been used in the livestock sector for three purposes:-

i.  to try to understand the structure of the chain to assess whether it is operating effectively and efficiently, from the perspective of the various interested or involved actors;
    Examples for (i) for pig meat include [11], [12], [13] and [14] whose common conclusion was that the pig meat supply chain in each case was sensitive to the actions, decisions and policies of major supermarkets, especially in terms of prices charged, cuts of meat marketed and supplier types stocked.

ii.  to calculate financial effects on actors in the chain, usually farmers and consumers, of improvements in animal health and welfare; and

iii.  to assess the consequences on the chain of a 'shock' to the system.

[15] provided a detailed treatment of how to calculate the production and financial effects of improving animal disease risk management. Specific country examples in this context include research provided by [16] and [17] which both showed significant improvements in financial performance by improving management of animal disease outbreaks.

[10] proposed a new framing of the problem in terms of opportunistic dealing adopted by supermarkets in vertically disintegrated supply chains, where all actors attempt to pass the risks and costs onto somebody else. [7] used VCA to analyse the effects of the veterinary surveillance system on the likelihood of disease outbreaks and the consequences of these. He demonstrated the usefulness of VCA in analysing animal health management and highlights the importance of information-sharing, collaboration and social capital and trust in veterinary surveillance. Here, we argue that trust and collaboration are also important in the context of production diseases.

VCA typically includes characterisation and appraisal of value chains and related information and design and evaluation of interventions to improve value chain performance [18]. Here, the start was analysing how specific intervention scenarios could be perceived within the value chain. These scenarios were derived from work during the PROHEALTH project, and are presented from the pig farming perspective in the 'Description of scenarios' section below.

Our analysis included the following steps (Fig 1):

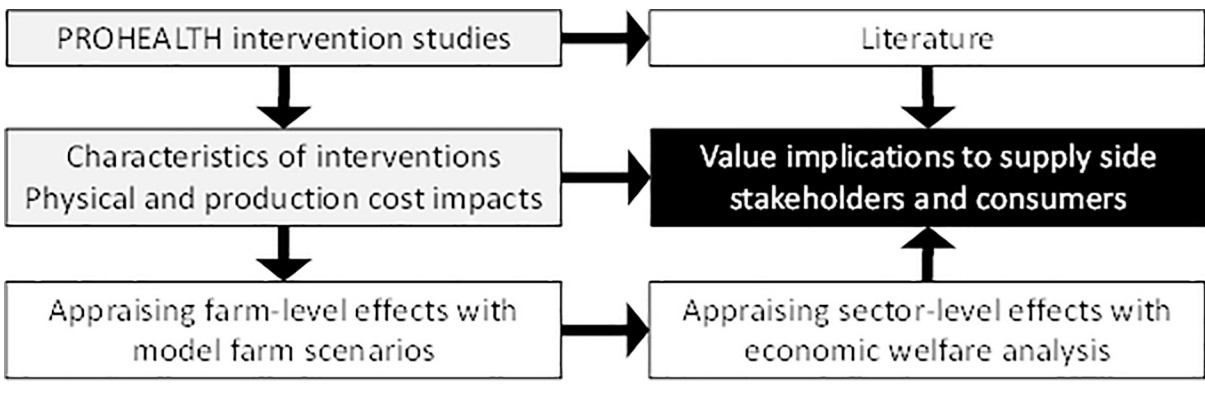

**Fig 1. Steps taken in the current analysis.**

1. Identifying effect of interventions to different stakeholder groups, and their views towards interventions, based on literature and previously conducted intervention studies;

2. Appraising financial effects of the intervention at the farm level; and

3. Appraising the effects of interventions at the markets.

Farm-level effects of each intervention per animal were described using information on the effects of interventions on production parameters, namely, piglet mortality, average daily gain of fattening pigs, feed intake per fattening pig and disease incidence per fattening pig and piglet. These data were obtained from the background studies summarized in Tables 1 and 2.

Following a literature review, a value chain diagram showing linkages between stakeholders and value-added dimensions was designed for the pig meat chain. We identified possible business and value consequences to stakeholders involved in the chain. The roles of the most important stakeholders in the chain, in relation to interventions studied, were described, and possible implications discussed.

To put the intervention scenarios into perspective, value chains in the UK and Finland, two very different countries, illustrated the financial effects of these interventions. During 2014–2017, the UK was the ninth largest pig meat producer in the EU and Finland was the 17[th] largest pig meat producer of the 28 EU member states [1]. Key information on performance, structure and size of pig value chains for the two countries was retrieved and is discussed below, with the information presented in subsequent sections. Information on how the value is distributed through the meat sector was also retrieved where it was available. The effects of interventions were then described for an average-sized pig farm and at the sector-level to illustrate their impact in the supply chain.

## Description of intervention scenarios

**Intervention 1: Improved hygiene in pig fattening.** Good hygiene is crucial in pig fattening. However, there are substantial differences in biosecurity practices between pig farms so there are possibilities for improvement [2,19]. Not all farms apply systematic and thorough daily cleaning and visitor biosecurity practices and a cleaning and sanitation period between batches of animals; [4] suggest that some 10–15% did not have a proper cleaning break between batches. Inadequate hygiene can lead to elevated incidence of disease, mortality and carcass condemnations, reduced animal welfare and lower financial returns.

This intervention scenario was based on a pig rearing trial by [20] which showed that dirty housing and inadequate biosecurity are risk factors for pig health, and that an intervention where hygiene practice was improved, enhanced pig performance and health. Here, a change from very dirty to very clean housing conditions was examined. Pigs kept in clean conditions were assumed to gain more weight than pigs kept in dirty housing. This implies that more

**Table 1. The mean change in production factors resulting from improved pig housing cleanliness.**

| Parameter[a] | Change |
|---|---|
| Mean body weight on the 85[th] day of fattening (kg) | +11 kg (13%) |
| Feed conversion ratio until day 85 (kg feed/kg gain) | -9.7% |
| Pleurisy (% slaughtered pigs) | -16% |
| Amount of meat produced (kg/pig space/year) | +38 kg (22%) |

[a]Information in this table is based on [20] and authors' calculations.

**Table 2. Mean change in production parameters resulting from interventions contributing to piglet viability.**

| | Change |
|---|---|
| **Measures to reduce piglet mortality** | |
| Impacts of thermoregulation on neonatal piglet mortality[c] | -2.4%[g] |
| Impacts of emphasizing disease robustness in genetic selection on neonatal mortality[d] | -2.6%** |
| Impacts of emphasizing disease robustness in genetic selection on rearing mortality[d] | -2%[o] |
| **Positive handling of sows** | |
| Pre-weaning mortality (%[e]) | -2.2%[h] |
| Days treated with antimicrobials, measured in the nursery[e] | -2.4[i] |
| Improvement of welfare, measured in the nursery by an animal welfare score[a,e] | -14% |
| **A high threonine-tryptophan feed (in comparison with high lysine feed) provided to problem-piglets** | |
| Feed conversion ratio (kg feed/kg gain) during the prestarter period for piglets from primiparous sows[f] | +10%* |
| Feed conversion ratio (kg feed/kg gain) during the starter feeding for piglets from primiparous sows[f] | +22%[b] |
| Piglets' weight gain during the prestarter period (kg/d[f]) | +55 g/d NS |
| Piglets' weight gain during the starter period (kg/d[f]) | -21 g/d NS |
| The price of pre-starter piglet feed[f] | +20% |
| The price of starter piglet feed[f] | +17% |

Statistical significance of change for parameters for which information was available

***p<0.0001

**p<0.01

*p<0.05

[o]p<0.1.

[NS] The effect of diet was statistically insignificant, p>0.1.

[a]Lower score indicates better animal welfare rating.

[b]The effect of diet was statistically significant only as a trend, at risk level 0.05<p≤0.1

[c][22].

[d][4]

[e]Based on an experiment described by [25].

[f]Based on an experiment described by [24].

[g] Coefficient of variation for mortality before the change was 0.28.

[h] Coefficient of variation for mortality before the change was 0.65.

[i] Coefficient of variation for the number of days before the change was 0.40.

output can be produced a year under clean conditions. The clean group also had less weight variation at slaughter than the dirty group and fewer respiratory lesions. As a result, the intervention was calculated to increase the amount of meat produced per pig space unit per year by about 22% (Table 1). The coefficient of variation for the difference of mass of meat at a given point of time was 37%. The occurrence of pleurisy and pericarditis was, in general, associated with reduced performance of the pigs.

In summary, there are benefits associated with the maintenance of good hygiene, and these come especially in: increased growth rate of pigs; improved homogeneity of carcasses; lower disease incidence; and reduced need to use medicines. These contribute to improved financial return (see Table 3).

**Intervention 2: Enhanced care of piglets and sows to improve piglet survival.** Piglet mortality and challenges related to piglet performance are important in pig production. Piglet mortality during parturition and lactation currently results in the loss of 15–20% of all piglets born, which increases with litter size [21]. This is of concern for both animal welfare and the

**Table 3. Key figures representing the pig production sector in Finland and the UK, 2017: Parameters characterising the market situation in the welfare analysis before applying an intervention.**

| Parameter | Finland | UK |
|---|---|---|
| Variable costs of pig meat production, €/tonne pig meat[a] | 1,280 | 1,560 |
| Fixed costs of pig meat production, €/tonne pig meat[a] | 290 | 230 |
| Producer price of pig meat, €/tonne pig meat[a] | 1,570 | 1,790 |
| Net benefit of 'improved pig housing cleanliness', €/tonne pig meat[b] | 189 | 216 |
| Net benefit of 'measures to reduce piglet mortality', €/tonne pig meat[b] | 5 | 8 |
| Net benefit of 'positive handling of sows', €/tonne pig meat[b] | 10 | 13 |
| Net benefit of 'a high threonine-tryptophan feed provided to problem-piglets', €/ton pig meat[b] | -6 | -30 |
| Quantity produced ($Q_b$), million kg pig meat[c] | 182 | 901 |
| Total costs, € million | 286 | 1613 |
| Producer price % of the retail price[d] | 26.4% | 35.0% |
| Consumer price ($P_b$), €/tonne | 5,606 | 5,314 |
| Own-price elasticity estimate for demand ($\theta$)[e] | -0.690 | -0.779 |
| Domestic supply, million kg | 182[f] | 901[i] |
| Number of slaughtered animals per year, million pigs | 2.0[f] | 10.65[i] |
| Exports, million kg | 32.3[h] | 228[i] |
| Exports, % of domestic production | 18 | 25 |
| Domestic consumption, million kg | 184[h] | 1,713[i] |
| Imports, million kg | 32.6[h] | 524[i] |
| Imports, % of domestic consumption | 18 | 46 |
| Number of premises with pigs | 1,160[h] | NA |
| Number of pig farms | 600[h] | 11,500[i] |
| Number of approved slaughterhouses | 38[g] | 130[i] |
| Approximate number of veterinarians working with pig farms | 300 | NA |

[a]Costs structure is based on [36] and is assumed to represent costs before adopting an intervention. Producer price refers to assumed market-clearing producer price of pig meat before an intervention has been adopted.
[b]Estimates obtained from [34].
[c][41,44].
[d][41,43].
[e][32,33].
[f][44].
[g][45].
[h][39].
[i]UK sources taken from the text.

financial sustainability of farm businesses. In addition, weak piglets need more intensive care to survive well.

This intervention was based on the PROHEALTH project's studies and included a set of measures to improve piglet survival. First, improved survival was assumed to be achieved by using gilts with superior genetics in relation to disease resistance or by providing piglets support for both suckling and thermoregulation [4,22]. Second, positive handling of sows was also assumed to reduce piglet mortality (based on a study similar to [23]. Third, adjustments in the dietary composition of feed fed to problem piglets were also considered [24].

The assumed effects of interventions on production parameters used in the analysis, are summarized in Table 2. [4] found a 2.6% decrease in piglet mortality at birth and a 2% decrease in pre-weaning mortality on farms which prioritized genetic selection for robustness (tolerance against multiple stressors, such as pathogens and environmental stressors) in their

replacement gilt program, when compared to farms which did not prioritize the importance of genetic selection for disease resistance. [22] reported that the total early-life piglet mortality was 2.4% lower in farms which provided regular help to piglets and several sources of supplementary heating than farms which provided only some assistance and limited support for suckling and thermoregulation. The former farms also had fewer piglets dying from starvation and crushing, or fewer non-viable piglets. In addition, positive handling of sows which included playing them music and backscratching daily for 15 seconds while they were in the farrowing pen was examined. Such positive handling appeared to reduce piglet mortality by 2.2%, based on the findings of [25].

The nutritional intervention component targeted on providing extra care to piglets at risk of showing poor performance and health. The amino acid feeding adjustment during the nursery phase could support the growth and health of pigs with higher risk of sickness. In this intervention, there are two options for feeding piglets from primiparous sows: a high lysine diet and a high threonine-tryptophan diet. Because tryptophan and threonine are involved in biological functions such as immunity, providing a high tryptophan-threonine diet instead of high lysine diet, normally used to maximize muscle tissue growth, could increase the growth of 'problem-piglets'. However, in the PROHEALTH experiment, the dietary treatment did not affect daily weight gain or average daily feed intake of pigs. By contrast, there was a tendency towards a higher FCR when applying the high threonine-tryptophan diet instead of high lysine diet. In addition, a kilogram of high threonine-tryptophan feed was 17–20% more expensive than a kilogram of high lysine feed.

There were limited benefits from the high tryptophan-threonine diet: although poor performance of problem pigs was found to be improved only in the immediate post-weaning phase with a better-suited feed, after the pre-starter phase, higher growth rate promoted by the enriched feed was mainly associated with higher feed intake, increased feed costs and a reduced financial return from pig production.

This measure affects the costs of work i.e. labour (scratching) and playing music. In addition, because of reduced mortality, more meat will be produced per sow. In summary, positive handling and selection paying attention to piglet mortality result in: lower piglet mortality thanks to the treatments; less days treated with antimicrobials; more sustainable production and better animal welfare scores, with greater involvement by handlers potentially increasing positive relationships with pigs.

## An economic approach to adopting an intervention

Based on an economic welfare analysis (see S1 Appendix for further detail), three economic reasons as to why an intervention might be implemented are particularly relevant in our case:

i. an intervention is adopted because it reduces production costs per unit of output leading to economic gains, without affecting demand for the product at a given price;

ii. an intervention is adopted because it increases demand for the product. This is associated with changes in product characteristics and consumers' preferences i.e. consumers have a WTP a higher price for the same amount of product than before the intervention, because the intervention enhances the quality characteristics of the product. This type of adoption requires an increase in the production costs per unit of product, if any increase occurs, is smaller than the premium that consumers are WTP for the product with an intervention; and

iii. when neither of the above justifications are feasible, an intervention can be adopted because of policy measures or because of coordinated activities taken by supply- chain

stakeholders. Policy measures can include, for instance, financial support targeted towards those who apply the intervention, or enforcement through a regulation, when markets are not able to adopt measures preferred by society (e.g. because of externalities or market failure).

Reason i) above was analysed quantitatively by means of economic welfare analysis presented in the section on 'sector-level microeconomic analysis' below whereas reasons ii) and iii) were analysed qualitatively by means of stakeholder analysis presented in the following section.

## Supply side actors and consumer views regarding interventions

Reflections of a range of stakeholders in the pig sector to the interventions were investigated in two ways. First, a stakeholder consultation survey in 2016 to around 300 experts in the pig sector in Finland, Germany, Spain, Poland and the UK; 100 replied. They were asked to indicate which of a range of interventions they considered the most appropriate to control production diseases in pigs and the extent they thought these would be successful. Second, a further consultation was carried out in October 2018 with the 34 original respondents from Finland and the UK to ask how they viewed the interventions presented here and what they thought might be the rates of uptake of these. The consultations were conducted by e-mail and hard-copy post.

Consumer views on interventions to reduce production diseases came from two systematic reviews [26,27] and a citizen survey conducted in 2017 [28] as part of the PROHEALTH project. The systematic reviews analysed consumer attitudes towards farm animal welfare (FAW) with specific focus on production diseases in intensive systems. [26] examined attitudes in general to FAW in animal production systems, and [27] focused on consumers' willingness to pay (WTP) for FAW in general and, FAW in relation to a reduction of animal production diseases, in particular. The key findings of these studies were that naturalness and humane treatment were central to what was considered good welfare by consumers in the study countries. The prophylactic use of antibiotics was identified as a key concern. Results relevant to the evidence gap identified were collected by [28] who assessed public opinion in the study countries from 2,330 responses; the data was analysed by Kruskal-Wallis ANOVA, exploratory factor analysis and structural equation modelling.

We investigated whether European and international animal welfare labelling and certification schemes addressed the interventions investigated. For this, the requirements of 12 pig labelling schemes were studied. As some of these schemes have specific variations in their requirements, altogether 22 pig specifications were examined. The data for this were collected during the period August to December 2017 from the internet sites of the schemes. The following pig schemes were checked: AWA; Beter Leven; Bedre Dyrevelfærd; Certified Humane; Coop (Denmark); Coop (Switzerland); Global Animal Partnership; Initiative Tierwohl; Naturafarm Coop; Red Tractor; RSPCA Assured; and Tierschutzlabel.

## Sector-level microeconomic analysis

**Economic welfare analysis.** Besides firm-level effects, we assessed the effects of interventions at the market level. A sector-level approach has been used to address the market effects of animal diseases by, amongst others, [29], [30] and [31]. Here, market-level effects of an intervention were quantified using a comparative static economic welfare analysis.

The economic effects of an intervention at the animal production sector depend on how markets respond to it. When an intervention affects production costs, and affects a substantial

proportion of production, it can influence price and quantity of meat traded on markets. In our analysis (which is elaborated in more detail in S1 Appendix), market-level welfare effects were calculated for producers and consumers separately and then jointly, which was assumed to represent society's welfare loss or gain from adopting an intervention. They included the effects on farms which had already adopted an intervention, as well as the effect on those who had not yet adopted it but would do so in future. Change in producer surplus ($\Delta$PS), and in consumer surplus ($\Delta$PS), was approximated by the following equations:

$$\Delta\text{PS} = Q_b(P_a - P_b) - Q_b(C_a - C_b) + (Q_a - Q_b)(P_a - C_V),$$

$$\Delta\text{CS} = Q_b(P_a - P_b) + (Q_a - Q_b)(P_a - P_b)1/2,$$

Subject to   $P_a \geq C_a$,  $P_b \geq C_b$ and market clearing,

$$Q_a = Q_b\left(1 + \frac{\theta(P_a - P_b)}{1/2(P_a + P_b)}\right).$$

$Q_b$ and $Q_a$ above are quantities traded before (subscript $b$), and after (subscript $a$), adopting an intervention, $P_b$ and $P_a$ are prices of pig meat before, and after, adopting the intervention, $C_b$ and $C_a$ are production costs of pig meat before, and after, adopting the intervention and $C_V$ represents variable costs for producing the additional amount $Q_a$-$Q_b$, and $\theta$ is own-price elasticity estimate for demand for pig meat, respectively. A market clearing is assumed which implies that the quantity supplied and quantity demanded equal and that prices demanded and offered in the market clearing also equal. The parameter values ($P_b$, $Q_b$, $C_b$) for the situation before adopting an intervention (i.e. in current market situation) as well as own-price elasticity estimate $\theta$ for pig meat are provided in Table 3.

In the model, adjustments in market prices are determined by how an intervention influences production costs whereas adjustments in quantity traded are determined by demand adjusting to the new price level as explained below. The effects of intervention on production costs and prices were defined as euro per tonne of pig meat and the industry was considered to operate with a fixed margin. The effects were then scaled up to represent the total volume of meat produced in Finland and the UK. It was considered for each intervention that the markets are competitive and the price difference $P_a$-$P_b$ is determined by fully passing the producer side efficiency gains (or losses) of interventions described in Tables 1 and 2 on to the consumer prices. Hence, the supply curve, representing the costs of production, would shift by the amount indicated by the cost change associated with each intervention. Price difference $P_a$-$P_b$ therefore corresponds to this net benefit or net loss and the producers' collective benefit or loss from an intervention is because of adjustments in the quantity supplied. These net benefits are provided in Table 3.

The farm-level net benefit ($\Delta\pi$) of an intervention was assessed as follows:

$$\Delta\pi = \pi_b - \pi_a = q_b p_b - \mathbf{x_b}\mathbf{w} - C_F - (q_a p_a - \mathbf{x_a}\mathbf{w} - C_F)$$

$$q_a = q_b + \Delta q_i$$

$$\mathbf{x_a} = \mathbf{x_b} + \Delta\mathbf{x_i}$$

Where subscript $a$ indicates situation after and subscript $b$ before adopting an intervention, $\pi_b$ and $\pi_a$ are net profits received by the farm, $q_b$ and $q_a$ are output quantities produced by the farm, $p_b$ and $p_a$ are output prices received by the farm, $\mathbf{x_b}$ and $\mathbf{x_a}$ are vectors of inputs used, $\mathbf{w}$ is vector of input prices and $C_F$ is the fixed cost, which is the same before and after adopting an intervention. $\Delta q_i$ represents change in the farm's output quantity as a consequence of intervention $i$, because the interventions may influence pig meat and piglet yields through impacts they have on pig mortality and growth rate, as explained upon description of interventions. $\Delta x_i(.)$

represents impacts of intervention *i* on input use and includes factors such additional labour input because of intervention and saved labour due to reduced disease, mortality-related costs, changes in antimicrobials and other inputs needed to treat production diseases, changes in feed consumption and other miscellaneous costs. Moreover, producers were assumed to be able to increase current production without using additional fixed inputs because efficiency gains of interventions studied here were associated with increased efficiency of using fixed inputs (e.g. Intervention 1 increasing pig meat output per pig space per year due to improved pig performance).

After an intervention had been implemented, the price asked by the sellers was assumed to change as deemed by net efficiency gains or losses. At the sector-level, the quantity of meat traded was assumed to adjust along the demand curve to correspond to the new price level. For each intervention, the change in the quantity of meat demanded was calculated by using own-price elasticity estimates from the literature and changes $P_a$-$P_b$ described above. Change in demand was quantified by elasticity estimates provided in Table 3 for Finland and the UK. These were based on almost ideal demand systems estimated by [32] and [33].

**Data sources for quantitative analysis.** The net benefits of interventions per animal were obtained from [34] and are provided in Table 3 for both countries. In the source study, the impacts concerning the pig fattening phase were assessed in the model similar to [35] and the impacts concerning piglets and sows were assessed in the model presented by [3]. Because the source did not account for farm-level effects, model farm scenarios were developed in this study to illustrate the impact of interventions for an average-sized pig farm in both countries. Production cost structure information used in the analysis came from Interpig reports [36] for 2015. The costs are representative for fattening pig farms, which means that the benefits from piglet-related interventions are assumed to be passed onto the pig fattening phase. This approach was chosen to have the results between interventions comparable with each other. However, the effects for an average-sized farrowing farm would be larger than the effects scaled for an average-size fattening pig farm. Farm size was defined so that the revenues from pigs corresponded to the financial results for specialist pig farms participating in costings for the UK as shown by [37] and, in Finland, for a specialist farm as shown by [38]. The baseline model pig farm scenarios are in S2 Appendix and summarized also in Table 3 per tonne of pig meat.

Table 3 illustrates the size of the pig production sector in the two study countries as it was used as a reference for sector-wide effects; 2M pigs were slaughtered in Finland in 2017 and the production of pig meat was 181M kg, consumption was 184 million kg, of which 18% was imported meat, and the value of production at primary producer prices (€1.48/kg) was €268M [39]. The pigs consumed about 750M kg of feed, the main input in terms of costs [40]. In the UK, 10.65M pigs were slaughtered with 2,277 head exported (worth £100 000) and 508,000 head (worth £61 million) imported [41]. Altogether, 2,014M kg of concentrated pig feed was supplied [42] to the 11,500 UK pig holdings [41].

In Finland, primary producers received 23.4%, the food industry 37.5% and retailers 27.6% share of consumer price of pig meat in 2012. The remaining 11.5% were taxes [43]. Primary producers' share of retail price for pig meat in the UK in 2015 was 35.0%. Hence, with an average consumer price of €5.31 per kg, the producer would have received €1.86 per kg pig meat (£1.36) [41].

Different scenarios regarding the adoption rate of interventions by farmers are discussed. All analyses compared situations where all farms, and no farms, applied the interventions. Because some farms are currently applying the interventions, the results represent effects that can be obtained by farms who have already adopted the measure plus benefits achieved by other farms if they adopt the measures in the future.

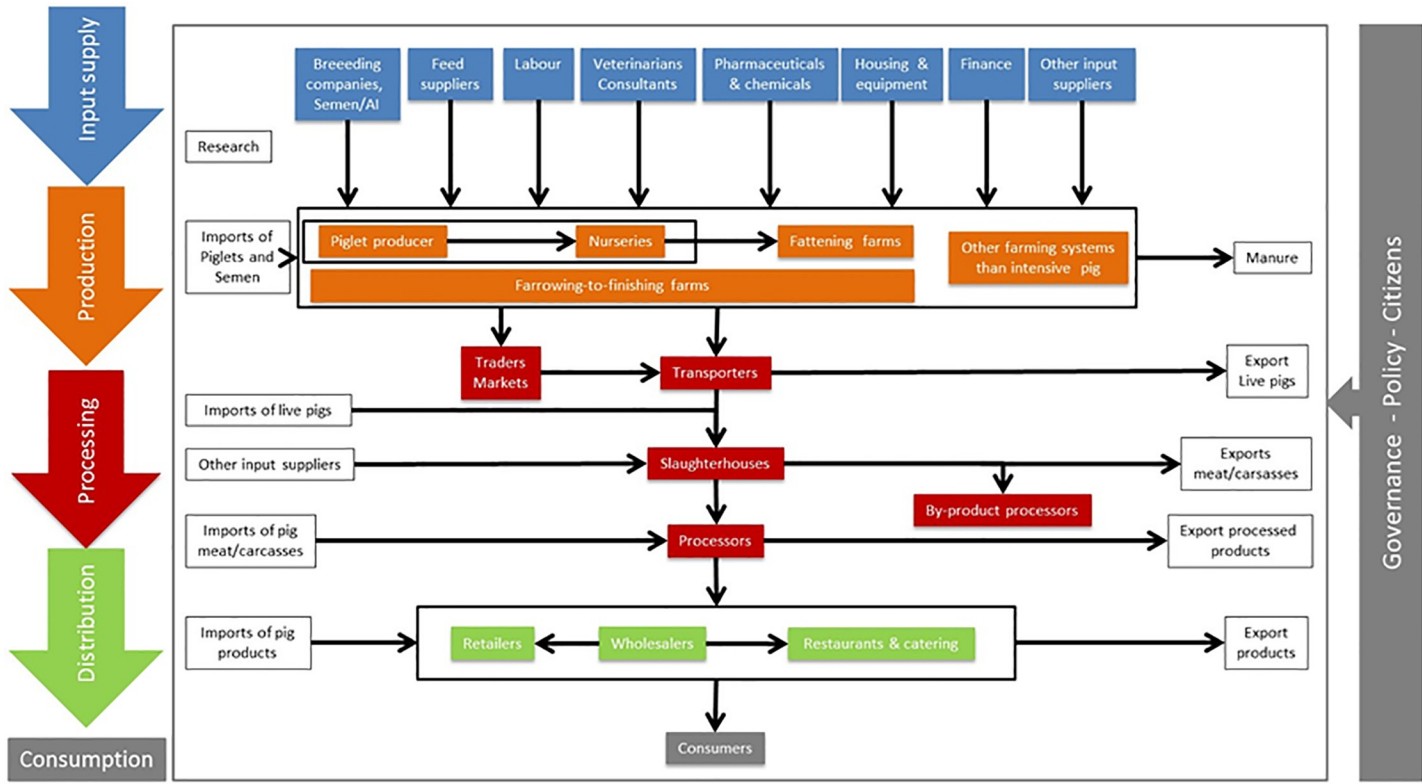

**Fig 2. A value chain diagram for intensive pig production.**

Table 3 represents values for 2017. A sensitivity analysis was carried out by increasing parameter values by 10% *(ceteris paribus)* from the values indicated in Table 3 and then comparing the results with those obtained before increase.

## Results

### Pig production value chain

**Characterisation of the pig production value chain.**   Fig 2 illustrates the main stakeholders involved in the intensive pig production value chain. This comprises farms which produce piglets or pigs for slaughter (farrowing farms, farrowing to finishing farms or fattening farms), actors who supply animals, feed, advice and other inputs to them, slaughterhouses or middlemen who purchase animals from farms, animal logistics and transport companies, meat processing companies who operate after slaughter, wholesalers and retailers and caterers who supply pig meat to consumers. Money goes in the opposite direction i.e. prices paid for goods and services flow from consumers towards primary production after margin obtained by each stakeholder is subtracted from their receipts.

Table 4 summarizes the main impacts identified as to how stakeholders could benefit from the interventions considered. For example, Interventions 1 and 2 can directly benefit pig farms and slaughterhouses by improving the efficiency of primary production and delivering more meat to slaughterhouses and food sector companies more competitively. This can also help competitiveness in international markets. Moreover, Intervention 1 is expected to add value through reduced lesions at slaughter and Intervention 2 because it responds to animal welfare concerns the public has and, thus, it offers possibilities for quality assurance schemes.

**Table 4. Summary of the main business impacts that the two interventions considered could have on stakeholders along the pig value chain.**

| Stakeholders | 1) Improved hygiene in pig fattening | 2) Enhanced care and handling of sows and piglets |
|---|---|---|
| Breeding companies | Improved performance of pigs due to better controlled management | Market for robust pigs |
| Feed suppliers | Benefits if farmers are able to reduce feed costs | Market for novel feed products |
| Veterinarians | Selling more hygiene advice; less treatment of sick pigs | Selling more advice; less treatment of pigs |
| Pharmaceutical companies | Market health care protocols; less use of medicines | Market health care protocols; less use of medicines |
| Farm workers | Less work, more fluent work if protocols are followed | Additional work; better job satisfaction |
| Housing and equipment suppliers | Hygiene procedures taken into account in designing housing products; housing that is easier to clean | Products which support thermoregulation |
| Finance | Reduced risk to investment | Reduced risk to investment |
| Farms | Increased turnover and return on capital | Mostly increased turnover and return on capital |
| Transporters, logistics | Heavier pigs and more feed to be transported | More pigs to be transported |
| Slaughterhouses, traders, meat processors | More meat for processing at lower costs; improved hygiene quality; potential for less carcass condemnations | Higher quality of products; animal-friendly labelled products which have a premium |
| Retailers, wholesalers, catering services | Potential for reduced input price and increased food safety | Potential for animal-friendly labelled products which have a premium |
| Consumers | Potential for reduced food price and added value through food safety and less antimicrobials used | Potential for added value through animal-welfare related product attributes |

Intervention 1 offers potential business opportunities for veterinarians and biosecurity experts to provide advice, whereas Intervention 2 can offer business opportunities for animal breeders to sell more robust pigs, companies to sell products that support thermoregulation and feed companies to sell special feeds, if they can develop products that can be sold competitively. In addition, taxpayers will be affected if tax revenues or public expenditures are altered as a result.

**Specific characteristics of the pig production value chain in Finland.** The structure of the value chain has implications on the adoption of interventions. As shown in Table 3, there are 1,160 farms with pigs in Finland, but only 600 were specialist pig farms in 2017. Over half of fattening pigs are kept on units of more than 1,000 pigs [39]. There are 38 approved slaughterhouses for pigs in Finland [45]. However, the pig industry has a high level of vertical integration and concentration ratio, as three major companies procure over 95% of pig meat in Finland. Most farms have a production contract with one of these companies. Slaughterhouses tend to provide a price premium to farms which follow specific company procedures, usually including the genetics and management practices applied. Piglet trade and logistics are mainly coordinated by these companies or by farrowing farms and fattening pig farms selling and buying piglets with established contracts [46]. Veterinary services are provided mainly by veterinarians employed by 60 municipalities [3,47]. There are roughly 300 veterinarians working with pig farms [48]. Retailing is also very concentrated, as two major retailers have 81.7% of market share of grocery sales [49].

The high degree of integration and concentration provides opportunities to prevent pig production diseases through coordinated actions. As an indication of industry-wide, multi-stakeholder collaboration, the Finnish pig sector has established a unique pig herd health classification system (Sikava), which covers approximately 90% of farms and 97% of production. Besides daily care of pigs, herd health status of enrolled farms is followed by meat inspection and scheduled veterinarian visits (typically 4–6 times per farm per year). At every visit, the veterinarian systematically checks specific parameters such as housing, presence of diseases, mortality rates and pig health [50]. Moreover, pig farms report the use of medicines to Sikava. This platform can be used intensively to improve the control of production diseases in pig farms, implement good hygiene and increased animal welfare in a coordinated manner and principals (i.e. slaughterhouses) can provide economic incentives for farmers to do so.

**Specific characteristics of the pig production value chain in the UK.** In the UK, 92% of production comes from about 1,600 assured farms including 10 corporate companies which account for 35% of UK breeding sows. The UK is also unusual in that 40% of the herd is reared outdoors [51]. Outdoor rearing poses challenges to animal health control. The UK pig industry has a very high level of vertical integration with 35 businesses accounting for 50% of pig meat production. Virtually all prime pigs slaughtered move directly from farms to the abattoir [52]. [10] found that 65% of all pig meat sold was by the 7 largest supermarket groups. Also in the UK, the high degree of integration may provide novel opportunities for enhancing the prevention of pig production diseases and that initiatives taken by major companies may influence the sector as a whole.

However, [10] argue that this supply chain is a 'mess' with much opportunistic dealing by supermarkets in pig meat procurement. There are 130 slaughterhouses dealing with pigs, processing 901M kg pig meat (worth £1,638M) and exports of meat of 228M kg (worth £294M) [41]. In 2017, UK consumers ate 1,713M kg of pig meat, 26.2 kg per capita [41]. Quite a large share of this quantity was imported. While some 524M kg pig meat (worth £937M) and around 550M kg of pig meat products (worth £1,595M) were imported into the UK in 2017 (46% of such consumption), exports of processed pig products were only 36M kg (worth £97M) [41]. This suggests that the UK markets may be sensitive to substitute imported meat for domestically produced meat if the price of domestic products increases relative to imports. The substantial share of outdoor production suggests that UK consumers may be willing to consume premium products.

## Stakeholder views on interventions for pigs studied

The interventions mentioned most frequently by 34 original responding pig stakeholders in Finland and the UK, as preferred ways to control production diseases in pigs, included: re-designing housing; provision of play materials for pigs; improved ventilation control; reducing stocking density; vaccinating pigs; enhanced monitoring of pigs and housing; improved biosecurity and hygiene; and adjustments to feed composition. These were preferred by 80–100% of respondents.

Intervention 1, improved hygiene in pig fattening, is clearly an intervention well-perceived by the stakeholders. All experts who responded indicated improved biosecurity and hygiene as their preferred approach to controlling production diseases, and nearly all did so for improved ventilation control. Improved biosecurity and hygiene was also mentioned as an appropriate way to reduce pre-weaning piglet mortality, post-weaning diarrhoea and the porcine enteric disease complex. In addition, ensuring air quality and ventilation, part of Intervention 1, was indicated as appropriate in reducing tail and ear biting. In the follow-up stakeholder consultation, Intervention 1 was viewed as one that would probably be adopted by farmers.

Intervention 2, enhanced care of piglets and sows, was also seen favourably by stakeholders. Adjustments to feed composition were the most-preferred ways to control production diseases in general, by about 95% of the experts. About 65% of experts indicated improved animal breeding for more robust pigs as an appropriate way to reduce peri-parturient dysgalactia syndrome, but fewer considered it as appropriate in reducing stillbirths and piglet mortality. About 90% of experts indicated monitoring and good care for sows during farrowing was appropriate in reducing stillbirths and, almost all experts, indicated that providing piglets with good thermal environment was appropriate in reducing pre-weaning piglet mortality. Hence, the main characteristics of Intervention 2 were mostly viewed favourably by stakeholders. However, it was shown that the labour needed for backscratching may lower adoption rate and that feed price is crucial. Thus, a feed-based intervention that increases feeding costs may not be seen as a viable intervention from a farm business perspective.

## Demand pull: Consumer views towards controlling production diseases

Few past consumer studies related to production diseases. In the review of public attitudes to FAW by [26], no studies focused on production diseases, with only 26.2% of those included referring to them or antibiotic use. In most cases, this simply referred to an absence of disease or avoiding antibiotics, unless treatment of the animal is required. Many concerns raised in these studies related to naturalness and human food safety. For the WTP review [27], only 4 studies specifically examined the public's WTP for FAW related to production diseases: [53] examined disease and health in pigs and beef cows; [54] examined WTP for disease resistance in fish; [55] examined lower limits for broiler chickens failing health checks relating to foot pad lesions (and associated dermatitis); and [56] examined WTP for earlier disease detection in broilers. A further 10 studies examined WTP for animal products produced without antibiotics.

Consumer concerns related to animal production were identified in two PROHEALTH reviews [26,27] and in an EU survey [28]. Results suggest that the public have concerns over intensive production systems including, in relation to FAW, the lack of naturalness in these systems (including animals being able to perform natural behaviours), humane treatment of animals and the use of veterinary medicines, in particular the use of antimicrobials and associated antimicrobial resistance.

Given these concerns, it is unsurprising that the most preferred interventions in the EU survey [28], across all three animals surveyed (pigs, broilers and layers), were those that can be viewed as the most proactive, natural and least invasive, namely measures related to housing and hygiene. The least preferred interventions were those that were medicine-based, which raised concerns in relation to humane animal care and human food safety concerns.

This implies that the public are most likely to prefer interventions perceived as 'natural' and which prevent production diseases. This includes, for example, animal housing, linked to human interventions such as provision of clean housing, and which allow animals to exhibit behaviours within their 'natural behavioral ranges'. Concerns about negative impacts on the food chain relating to food safety were also expressed, particularly relating to veterinary pharmaceutical interventions, including antibiotic usage, vaccination and the use of probiotics. At the same time, the reviews and citizen survey [26,27,28] suggest that the public do not really know what production diseases are and have limited knowledge about how the food they purchase is produced. This is linked to cognitive dissonance, whereby people reduce the stress they experience when thinking about production systems, and the process of food production with their own behaviours in relation to food choices involving meat consumption [57,58].

The public appreciate naturalness and the application of preventative approaches to control production diseases or to produce animal-based food. One implication is that precautionary hygiene measures are strongly preferred as a way of controlling production diseases. In this respect, Intervention 1 will be preferred by the public. Naturalness and humane treatment of animals are particularly important in relation to animal welfare. From this perspective, the positive handling intervention would also be viewed very positively, while the clean housing intervention could also result in the pigs being kept in a barren environment. The public regard antimicrobial use as a proxy for production diseases and they have some concerns about antimicrobial use and food safety [28]. This suggests that the public are 'anti-interventionist' regarding medication-based interventions, partly due to food safety worries. In particular, they dislike prophylactic use of antibiotics, but they also have reservations regarding the use of vaccines. This view favours the two interventions considered as none of them are based on medication.

Few trusted information provided by the food industry. The most trusted organizations included: animal health; consumer organizations; quality assurance schemes; and government

bodies. This highlights the need to develop and transmit collaborative messages from trusted bodies [28]. External accreditation and assurance would, therefore, act as a guarantee to trust on food safety, animal health and wellbeing. Coordinated communication from the whole food system would ensure public trust.

Mechanisms for improving trust in the food industry might usefully include consideration of what is important to the public. In this respect, adopting interventions considered more natural and more proactive is important and including them in communication is essential. People in general have a negative attitude towards modern farming, because it is considered to breach 'naturalness'. Consumers associate higher welfare with other positive product attributes: safety, healthiness, quality, environmental attributes. Hence, these should be developed jointly to ensure trust towards animal production.

[26] found younger participants, and those with higher levels of education, showed concerns about FAW relating to production diseases and were more likely to pay a premium for FAW products. From the consumers' perspective, naturalness needs to be defined through animals having appropriate housing, space and access to resources. For consumers, it is important that biological needs are met and that behavioural characteristics are taken into account. Animal psychological health is also seen as important to consumers so the positive handling intervention is likely to be viewed positively.

Table 4 suggested labelling as one avenue to advance the adoption of Intervention 2. Labelling is the preferred way of conveying information about production systems to the public. In market-based solutions, there is need to emphasize the benefits of different interventions, particularly those that align with consumer priorities and preferences. This implies that attributes which are important to consumers should be communicated in market-based approaches.

There is a small positive WTP for improved FAW as estimates for pig products tend to be significantly lower than for dairy products and eggs [27]. However, production diseases represent a different consumer issue which overlaps FAW. There is also little information on consumers' WTP for a reduction in production diseases in farm livestock.

There are national variations in consumer attitudes to FAW, their concerns and WTP for animal welfare. The literature which, is generally from Europe and North America, indicates that in Northern Europe, where countries tend to have stricter regulations, WTP for animal welfare was lower than in Southern Europe [59], [60]. Consumers in Northern Europe pay more attention to legislative solutions and, in Southern Europe, more to market-based solutions. This may imply that there is more room for market-based solutions and quality assurance schemes to control production diseases in Southern Europe than in Northern Europe, but it may also imply the need to harmonize such policies across Europe.

Having higher welfare products is one means to ensure consumers can express preferences for products produced through different means and standards. One means of facilitating this, would be to ensure that welfare-friendly products, or products from different production systems, are clearly identifiable. Labelling appears to be the preferred consumer means for doing this, providing that there is a consistent and trustworthy labelling scheme in place, probably developed through multi-stakeholder involvement, including those independent of production, to ensure the credibility of the labelling schemes in place [26].

These findings provide insight regarding how the public could view the two interventions we investigated. These aspects are summarized in S3 Appendix. Intervention 1 has characteristics suggesting that it is not a medication-based preventive intervention; it is seen as an essential and acceptable intervention amongst the public. Intervention 1 can, also, help to build public trust towards livestock production, as it tackles concerns related to both animal health and food safety. However, this intervention may be seen as quite 'technical'. Intervention 2 addresses animal psychology aspects and positive welfare. Considering it as a humane

intervention, it could be considered as addressing consumer concerns well. Positive handling could also be incorporated into welfare labelling systems or standards, and has the potential for obtaining a price premium from consumers.

## Financial and economic impacts: Illustration for the UK and Finland

**Implications of interventions at the farm-level.** Table 5 represents simulated financial impacts of interventions for an average-sized pig fattening farm which adopts the intervention measures in full and has not applied them earlier. The impacts are larger for the improved hygiene interventions than for the other measures because it has the potential to result in larger net benefits. Improved hygiene in pig fattening can lead to a substantial increase in the gross margin and turnover of a pig farm which has not adopted the measure previously. Moreover, larger impacts were calculated for the UK model farm than that in Finland, mainly because of the larger farm size in our analysis. Measures to reduce piglet mortality and positive handling of sows were estimated to have the potential to increase pig farm gross margin by 1.5 to 3.3% whereas the piglet feeding measure had a negative impact on gross margin. In the case of the piglet feeding measure, a subsidy that compensates producers for the additional cost incurred, would restore supply to the original level.

**Implications in the pig supply chain.** Passing the costs and benefits to the consumer price of pig meat in full, the improved pig hygiene intervention was found to have the potential to reduce the consumer price of pig meat by up to 5%. For other measures, the impact was less than 0.5% (Table 6). The industry was considered to operate with a fixed margin implying that the percentage change in producer prices is larger than that in consumer prices.

From our assumptions, it follows that a reduction in the variable production costs is transmitted to the consumer price, and lowering the price of meat is accompanied by a positive response in demand which partially compensates for the producers' income loss due to the producer price reduction. This response depends on the price elasticity of demand and it was at most 3.9% for Intervention 1 in the UK and 2.6% for Finland. For other measures, adjustments in quantity were at most 0.2%.

**Implications of pig interventions to the value of production.** Table 7 shows estimated economic welfare changes i.e. changes in producer and consumer surplus, and these together

**Table 5. Financial net impact of interventions (€/farm/year) in Finland and the UK when the effects of interventions are passed on to a fattening pig farm.**

| Intervention | Finland | UK[a] |
|---|---|---|
| Improved hygiene in pig fattening | 38 516 | 70 620 |
| % of gross margin | 49.2 | 51.6 |
| % of turnover | 12.3 | 13.3 |
| Enhanced care and handling | | |
| -measures to reduce piglet mortality | 1 167 | 2 816 |
| % of gross margin | 1.5 | 2.1 |
| % of turnover | 0.0 | 0.0 |
| -positive handling of sows | 2 333 | 4 576 |
| % of gross margin | 3.0 | 3.3 |
| % of turnover | 0.0 | 0.0 |
| -targeted piglet feeding | -1 400 | -3 210 |
| % of gross margin | -1.8 | -2.3 |
| % of turnover | 0.0 | 0.0 |

[a] €1 = £0.7258

**Table 6. Estimated impact of the interventions price parameters (% from before intervention parameter) as a consequence of adopting an intervention in all herds (100% national supply affected).**

| Parameter | Finland | UK |
|---|---|---|
| *After improving hygiene in all herds* | | |
| Change in variable costs, €/tonne | -16,3% | -17,1% |
| Change in producer price, % | -13,3% | -14,9% |
| Change in consumer price | -3,7% | -5,0% |
| Change in quantity traded | 2,6% | 3,9% |
| *After measures to reduce piglet mortality in all herds* | | |
| Change in variable costs, €/tonne | -0,4% | -0,6% |
| Change in producer price, % | -0,4% | -0,6% |
| Change in consumer price | -0,1% | -0,2% |
| Change in quantity traded | 0,1% | 0,1% |
| *After positive handling in all herds* | | |
| Change in variable costs, €/tonne | -0,9% | -1,0% |
| Change in producer price, % | -0,7% | -0,9% |
| Change in consumer price | -0,2% | -0,3% |
| Change in quantity traded | 0,1% | 0,2% |
| *After improving piglet feeding in all herds* | | |
| Change in variable costs, €/tonne | 0,5% | 0,7% |
| Change in producer price, % | 0,4% | 0,6% |
| Change in consumer price | 0,1% | 0,2% |
| Change in quantity traded | -0,1% | -0,2% |

in Finland and the UK when quantities traded and prices observed in 2015 are used as a baseline. The estimates include benefits estimated on farms who have already adopted the measure, and benefits achieved by other farms if they adopt the measure. The total welfare benefit if none of the pig farms had adopted the improved hygiene intervention, but would adopt it now, was estimated at €37.5M per year for Finland and €235.5 for the UK. The total welfare benefit from the hygiene intervention, if all farms had already adopted it, was estimated at €39.9 million per year for Finland and €255.1 for the UK. These numbers represent the benefits for all farms. Obviously, no further benefits could be obtained if all farms had already adopted good hygiene practices and, hence, the results of that case indicate the benefits that would have already been obtained. In practice, if 10% of current farms had the capacity to

**Table 7. Estimated welfare effects of the studied interventions on pigs in Finland and the UK when year 2015 quantities are used as the starting point (farms which have currently adopted interventions and those who would adopt them in the future) €M/yr.**

| | Consumer surplus[a] | | Producer surplus | | Net welfare change[a] | |
|---|---|---|---|---|---|---|
| | Finland | UK | Finland | UK | Finland | UK |
| **Improved pig hygiene** | **37.5–38.5** | **235.5–244.8** | **1.4** | **10.2** | **37.5–39.9** | **235.5–255.1** |
| Enhanced care of piglets and sows: | | | | | | |
| Reduced piglet mortality | 1.0 | 8.9 | 0.0 | 0.4 | 1.0 | 8.9–9.3 |
| Positive handling | 2.0 | 14.5 | 0.1 | 0.6 | 2.0 | 14.4–15.1 |
| Improved piglet feeding | -1.2 | -10.1 | 0.0 | -0.4 | -1.2 | -10.2–10.6 |

[a]The consumer surplus and new welfare effect reported here include surplus to taxpayers, as a consequence of changes in Value Added Tax revenues. For Finland, the value of tax revenues was calculated at 22.9% of change in consumer surplus and 23.7% of the new welfare change. In the UK, the value-added tax of food is zero for most foodstuffs. The tax effect of services related to food supply chain are excluded from the analysis.

adopt the pig hygiene intervention, the benefits that could be obtained by the sector for consumers and taxpayers are approximately €4M in Finland and approximately at €25M in the UK.

In addition, benefits calculated per intervention are larger for farms which have already adopted the measures than for farms which haven't because adoption of an intervention was assumed to influence producer prices in markets. However, the pig hygiene measure had a substantial impact on the market clearing prices and quantities. Measures to reduce piglet mortality were estimated to provide welfare benefits worth about €3.0M in Finland and about €24M per year in the UK.

Table 7 indicates that when an intervention is adopted at large, the majority of benefits would be transmitted to consumers, in the form of more affordable products. Tax payers may also benefit to some extent.

In this Section we have only examined impacts obtainable because of improved production efficiency associated with the interventions, and because of impacts that changes in market productivity could have on quantities traded in the markets. Moreover, this Section is based on the assumption that the markets are competitive and that no premiums are obtained through product differentiation.

In a sensitivity analysis, the parameter values provided in Table 3 were increased by 10% *(ceteris paribus)* and then the economic welfare analysis results were compared with those obtained before the increase. Changes in the variable costs of pig farming had only a marginal impact on economic welfare when compared to the results reported in Table 7, whereas an increase in fixed costs raised producer surplus. An increase in the average slaughter weight resulted in an almost similar decrease in welfare benefits. A 10% increase either in the quantity of meat produced, the average slaughter weight, estimated per kg or per animal benefit or cost of an intervention, or own-price elasticity estimate for demand, resulted in an almost similar increase in welfare benefits.

**Welfare assurance schemes in relation to the interventions studied.** Our internet and document search indicated that there are various labels from certified animal-friendly production in Europe and internationally. There were several schemes for the UK market, but no schemes targeting explicitly pig meat markets in Finland at the time of study. However, in addition to animal welfare labels, there are programs targeting antibiotic-free meat products, which have recently gained importance in Europe (for example a Danish initiative and two programs in Finland).

https://www.globalmeatnews.com/Article/2018/03/08/Denmark-sets-antibiotic-free-pig-target

http://www.bioeconomy.fi/finnish-chickens-are-reared-without-antibiotics/

https://www.dsm.com/content/dam/dsm/anh/en_US/documents/Eubiotics_Whitepaper.pdf

All pig welfare schemes examined had some requirement for animal health, but these varied. At least four pig schemes explicitly regulated practices related to antimicrobial use. Hygiene and hygienic monitoring practices, or piglet nutrition, were not addressed explicitly by these welfare assurance schemes. However, they are often addressed by company best practice guidelines. At least one pig scheme addressed the cleanliness of bedding and enrichments to the environment.

At least 15 schemes had guidelines regarding the weaning age of piglets, and 19 schemes had guidelines regarding the pen design or space allowance for the sow which, in some cases, included restrictions on the use of crates. Only two schemes were found to have a requirement regarding scratching the sows.

## Discussion

Here, we report on work that examined two broad intervention scenarios to reduce production diseases in pigs. The analysis suggests that these can add value in different ways. First, they can improve production efficiency and reduce production costs, as is the case in measures to improve hygiene in pig fattening and to reduce piglet mortality. This is the case for good hygiene, as it can reduce losses caused by production diseases and the level of antimicrobial use. On individual farms, these measures contribute positively to financial margin. If these interventions are implemented widely on farms, benefits can be passed onto consumers and, hence, food prices will be reduced. These results regarding market-wide effects are in line with [30], who shows how a respiratory disease diagnosed in the pig production chain can lead to loss of economic surplus by consumers and producers, and [61] who illustrate that changes in health management can lead to a series of adjustments, with potential repercussions for prices and volumes through livestock markets. Because most pig farms have adopted this intervention, at least partially, only a proportion of the estimated full impact remains to be achieved. While some farms could still achieve full benefits, others could achieve smaller gains by improving their hygiene. Expert advisers can use this result to show farmers the importance of good hygiene.

Second, the studied interventions can provide new business opportunities for livestock farms, slaughterhouses, food sector companies and other stakeholders as indicated above. To illustrate some opportunities: food sector companies can develop quality assurance schemes which highlight different characteristics of production systems; pig breeding companies have opportunities to develop more robust animals; animal health sector companies have opportunities to develop more holistic animal health care products paying attention to preventative measures; and, for manufacturers, there are opportunities to improve the design of housing and less costly pig diets.

In order to motivate livestock farmers to put additional effort into improving animal health, it is essential they have economic incentives to carry out the necessary investments. As shown by the interventions examined here, these can include investment in improved housing or genetics, and effort put into adjusting management procedures. It is particularly important for investments which involve the expenditure of costs over several years that upstream stakeholders, such as retailers, accept a financial responsibility and provide a commitment to compensate for these to the farms for instance in the form of a price premium paid on top of the regular product price when the farmer is participating in a certification scheme. Some of the animal welfare certification schemes reviewed for this study have such a feature.

Some interventions may require additional incentives from policy-makers, food companies or bodies coordinating animal health management if they are to be adopted. This observation coincides with the note of [62] for the dairy sector that the challenge facing the industry is to balance decisions and practices that may provide short-term economic gains, but may also be associated with long-term risks regarding sustainability. Hence, policy measures to incentivise this type of systemic intervention may be needed.

Third, the adoption of an intervention can be influenced by the value that it provides through quality characteristics of meat products. The public appreciate naturalness and the application of preventative approaches to control production diseases or, more generally, to produce animal-based food. They dislike medication-based interventions. In this respect, preferences of producers and the public do not conflict. Previously, [63] had noted that the interpretation of the complex and multi-dimensional concept of FAW can be quite compatible from a citizen and farmer perspective.

One of the implications is that precautionary hygiene measures are strongly preferred as a way of controlling production diseases. Each intervention we examined are preventative and,

thus, they can add value. Based on emerging antimicrobial-free production and existing animal welfare quality assurance schemes, there appears to be market potential for interventions which are considered animal-friendly and apply carefully targeted, need-based medication of animals. However, even when aiming at antimicrobial-free products, care should be taken that ill animals receive appropriate veterinary care because not providing good care to ill animals may compromise animal welfare. Positive handling of sows and reduced piglet mortality, and improved hygiene appear to be win-win interventions which are likely to add both value to the consumer and financial benefits to farmers although the farmers may consider additional value low. The positive handling intervention, which addresses animal psychological health, could be particularly important to consumers. These interventions could be incorporated into quality assurance schemes, thus providing new business opportunities to food sector operators. Strengthening the role of quality assurance schemes could reduce the stress people experience when thinking about animal production systems (see Section on consumer views above [57,58].

Changing emphasis from volume to value and increasing the market penetration of concept-based production, could also help to build public trust and how they value intensive livestock production systems. Improved hygiene, when followed by appropriate interventions, could also reduce worries related to food safety and antimicrobial use, which have been identified [26,27,28]. Policy interventions to promote these management changes can also be warranted because the risk of antimicrobial resistance to animal antimicrobial use and food safety are societally important issues. Although communication between producers and the public is essential, [64] noted that the public's education and exposure to livestock farming may resolve certain concerns, while other concerns will likely persist, especially when practices conflict with deeply held values around animal care.

In order to have livestock farmers put additional effort into improving animal health it is essential that they have financial incentives to carry out the required investment. As illustrated by the interventions examined here, these can include investment in improved housing or genetics, and efforts put on adjusting management procedures. It is particularly important for investments which carry costs over several years, that upstream stakeholders, such as retailers, provide a commitment to compensate the additional costs to the farms, for instance in the form of a price premium paid on top of the regular product price when the farmer themselves commits to a certification scheme. Some of the animal welfare certification schemes reviewed for the study discussed here have such a feature.

Economic factors are influential when farmers decide on the use of animal health management measures [2,65,66]. However, farmers' perceptions of the effectiveness of measures, availability of information [66,67] and ease of using measures [68] can impact health management. Economic and non-economic factors may also influence which interventions are recommended by veterinarians. Policy measures can be taken to promote interventions which are not financially viable *per se*, but which provide sufficient benefits in terms of animal welfare, food safety or public health, and may thus contribute a public good.

One measure appeared not viable without a policy intervention. The improved piglet feeding intervention had a positive effect on piglet performance during the immediate post-weaning phase, but financially it was affected by high costs of threonine-tryptophan feed. However, feed-based interventions in pigs are preferred by consumers, and piglet viability is an issue. As feeds can be designed in many ways, there is a need to develop low-cost feeds which could support the viability of problem-pigs in a targeted manner. Policies could also address animal nutrition. Policies could be designed also to boost interventions which rationalise medicine use because antimicrobial resistance is a growing concern to society. When policy interventions are needed, one option is that society introduces subsidies to help with increased costs, thus incentivising producers to adopt the desired interventions.

There are knowledge gaps regarding production diseases in both the natural and social sciences. There is a need for a more systematic approach to animal health interventions. As ways to control production disease can have conflicting impacts on various parameters, and positive and negative impacts to various stakeholders, these effects should be considered altogether. Hence, more evidence-based policy is required. As pointed out by [69], evidence-based decision-making is also needed to develop certification schemes.

In our analysis, facilitators of animal-friendly policies were also addressing public goods and governance options. The structure of the pig value chain also provides opportunities to control production diseases. The high degree of vertical integration implies that principals, such as slaughterhouses who process animals from farms, can have the potential to adopt best practices in a standardized and systematic manner. They can also use procurement contracts and company-level best practice guidelines to incentivise the adoption of interventions on farms. This approach could suit the best interventions which influence the disease pressure that the supply chain is facing or interventions which improve animal welfare (linked to a certification scheme) or reduce food safety risks. Both interventions studied here could, potentially, be enforced in this manner. In particular, this is an opportunity to adopt biosecurity and hygiene interventions, and positive handling practices.

Here, we focused entirely on intensive production systems. As we did not look at free-range or extensive production, or other species than pigs, these other animal systems need to be investigated separately in future.

## Conclusions

The public, and other stakeholders, appreciate preventative approaches to control production diseases or, more generally, to produce animal-based food. Our results indicate that good hygiene, robust animals and their positive handling as well as good management, to the extent addressed by the intervention scenarios, can add value to livestock value chains. However, interventions are not economically or societally preferred *per se*, because their financial and social viability is dependent on the characteristics of the interventions and the structure of the supply chain influences which policy measures can be applied. Because pig systems tend to be vertically integrated, this provides opportunities to adopt interventions which look at animal health from the system perspective.

More evidence is needed to support public policies, and business decision-making in this sector. Further research-based evidence on what are consumer attitudes regarding production diseases is essential for the sustainability of livestock production. In addition, financial analysis is necessary to identify case-by-case whether interventions not studied here are viable, and how added value would be distributed along the food chain. These knowledge gaps should be filled by further research.

## Supporting information

**S1 Appendix. Analytical example using economic welfare analysis.**
(DOCX)

**S2 Appendix. Baseline model farm scenario used in the analysis for pig meat, 2016/17.**
(DOCX)

**S3 Appendix. Summary of production and consumption side perspectives related to the characteristics of the two sets of interventions to control production diseases in pig production.**
(DOCX)

## Acknowledgments

Information provided by Nathalie Le Floc'h (INRA), Anna Stygar (Luke), Ilias Chantziaras (University of Ghent), Gema Montalvo and Carlos Pineiro (PigChamp Pro Europa s.l), Sotiris Papasolomontis (Vitatrace Nutrition Ltd), Ida Pedersen (University of Copenhagen), Dimitri de Meyer (Vedanko Bvba), Panagiotis Sakkas and Ilias Kyriazakis (Newcastly University) to help develop the intervention scenarios, is gratefully acknowledged by the authors. Teresa Hicks (University of Reading) played a major role in the mechanics of the stakeholder consultation on the studied interventions.

## Author Contributions

**Conceptualization:** Jarkko Niemi, Lynn Frewer, Philip Jones, Richard Tranter.

**Data curation:** Jarkko Niemi.

**Formal analysis:** Jarkko Niemi, Thomas Rimmler, Richard Tranter.

**Funding acquisition:** Jarkko Niemi, Richard Bennett, Lynn Frewer, Richard Tranter.

**Investigation:** Jarkko Niemi, Beth Clark, Thomas Rimmler, Richard Tranter.

**Methodology:** Jarkko Niemi, Richard Bennett, Beth Clark, Lynn Frewer, Philip Jones, Thomas Rimmler, Richard Tranter.

**Project administration:** Jarkko Niemi.

**Resources:** Jarkko Niemi.

**Supervision:** Jarkko Niemi, Lynn Frewer, Richard Tranter.

**Writing – original draft:** Jarkko Niemi, Beth Clark, Richard Tranter.

**Writing – review & editing:** Jarkko Niemi, Richard Bennett, Lynn Frewer, Philip Jones, Thomas Rimmler, Richard Tranter.

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
