## [Decision Letter · Decision Letter 0]

9 Jan 2020

PONE-D-19-30807

A value chain analysis of interventions to control production diseases in

the intensive pig production sector

PLOS ONE

Dear Professor Tranter,

Thank you for submitting your manuscript to PLOS ONE. After careful consideration, we feel that it has merit but it needs some revisions in order to fully meet PLOS ONE’s publication criteria. Therefore, we invite you to submit a revised version of the manuscript that addresses the points raised during the review process.

Specifically, I would recommend carefully taking into account the suggestions by the second reviewer and to address in the cover letter of the new submission the concerns raised by the first reviewer. 

We would appreciate receiving your revised manuscript by Feb 23 2020 11:59PM. To enhance the reproducibility of your results, we recommend that if applicable you deposit your laboratory protocols in protocols.io, where a protocol can be assigned its own identifier (DOI) such that it can be cited independently in the future. For instructions see: http://journals.plos.org/plosone/s/submission-guidelines#loc-laboratory-protocols

We look forward to receiving your revised manuscript.

Kind regards,

Nikolaos Georgantzis

Academic Editor

PLOS ONE

Journal Requirements:

Reviewers' comments:

Reviewer's Responses to Questions

**Comments to the Author**

1. Is the manuscript technically sound, and do the data support the conclusions?

Reviewer #1: Yes

Reviewer #2: Yes

2. Has the statistical analysis been performed appropriately and rigorously? 

Reviewer #1: Yes

Reviewer #2: Yes

3. Have the authors made all data underlying the findings in their manuscript fully available?

Reviewer #1: Yes

Reviewer #2: Yes

4. Is the manuscript presented in an intelligible fashion and written in standard English?

Reviewer #1: Yes

Reviewer #2: Yes

5. Review Comments to the Author

Reviewer #1: This paper reports the findings of a large EU funded project on the effects of two different interventions to improve pigs production. The study reports both the findings of the treatments, as well the results of the public perception of these interventions. This is an important study that can provide guidance to (i) producers by indicating the positive effect of simple interventions to improve pigs production, (ii) policy makers by highlighting that some simple interventions can benefit pig production within a good animal welfare environment. As the study involved both animals as well as humans, I would believe that some level of ethical approval was needed to carry out the different stages of the study. The authors claim that no ethical approval was needed, but I would suggest that the authors had obtained ethical approval and would be good to see evidence of that.

I would suggest the authors to check for any typos in the paper.

Reviewer #2: This is an interesting paper on a relevant topic for farm managers and policy makers, to analyse value effects of specific disease mitigation interventions in the pig production chain. The paper includes a thorough literature review. The methodology used (value chain analysis) is considered to be appropriate and results and conclusions are of relevance to farm managers and policy makers. Please find below a number of minor points/suggestions that could help improving the manuscript.

Minor comments:

1) Perhaps clarify/rephrase (abstract, lines 35-37): "However, interventions are not economically or societally preferred per se, because their financial and social viability is dependent on the interventions". Do you mean that an evaluation of the financial and social viability of the interventions is needed to recommend/decide what interventions are to be implemented?

2) Indent lines 89-92

3) Suggestion: I would suggest to mention the use of VCA in the introduction section and move the VCA lit review to the material and methods section (making it more succint). I would think this would focus the introduction to the problem in hand whilst touching on how to go about it avoinding dwelling too much on the methods used.

4) Description of intervention scenarios: Improved hygiene in pig fattening. Based on results from previous research the amount of meat produced per pig space unit per year is calculated to be 22%. Presumably this is an average/single case study. Do you have any information about the level of certainty of this figure (i.e. how much it could vary depending on circumstances)?

5) Table 2 shows change in production parameters resulting from interventions contributing to piglet viability. As above do you have any indication on how senstitive these parameters may be to circumstances? Providing some information on the level of certainty on the parameters used would be helpful for ascertain the defree of confidence of the results obtain. This does not mean that the results obtained here are irrelevant. On the contrary, I think results obtained are very informative for managers and policy makers, but by providing information (when possible) on the level of confidence on the parameters used would give further clarity of the results obtained.

6) Appendix 1 (lines 1300-1303): Perhaps clarify that you are referring a situation with intervention in place

7) Line 1316: Would it not be P1 instead of P0 (for D0)

8) Line 1320: "requires that the intervention increases..." Would it not be changes instead of increases? Market prices and quantities can change either with an increase or a decrease of production costs.

9) Line 1340 do you mean equilibrium (P0,Q0) instead of origin?

10) Lines 1355 and 1358: Would it be better to put Q1-Q0 instead of change in Q1

11) Line 367: Just chacking the last term of the change in producer surplus equation. Does it not need to be multiplied by 1/2?

12) Table 3 shows average figures. As above, would it be possible to say something about the distribution of these parameters? How would results be affected if these parameters change? I am not asking to conduct a sensitivity analysis (although you could do so), but to shed some light by commenting, if possible, on the sensitivity of results and implications of pig interventions to the value of production to variation in the parameter values used.

6. PLOS authors have the option to publish the peer review history of their article (what does this mean?). If published, this will include your full peer review and any attached files.

Reviewer #1: No

Reviewer #2: No

---

## [Author Response · Author response to Decision Letter 0]

5 Mar 2020

First, to address the concerns raised by Reviewer #1, namely the issue of ethical clearance of the research programme. The various pig experiments mentioned in the paper, had both ethical and welfare clearance by the EU’s DG Research Framework Programme 7 research assessment procedure before the research funds were released for the PROHEALTH project that this paper stems from. Furthermore, each of the Universities, Research Institutes and commercial farms that carried out the discussed experiments in the paper, had their own internal ethical clearance procedures approved. For example, the Intervention 1 on hygiene measures, was based on an experiment at the INRA facilities in compliance with the French directive on animal experimentation and care (2013-118) and was authorised from the French Ministry ethical committee No. 07.

Second, the pig experiments mentioned in References 24 and 25 did not require any additional ethical clearance as they were carried out on the owned commercial farms of the study participants. Similarly, data from References 4 and 22 did not require any additional ethical clearance in addition to that in the original application to the EU’s DG Research, as no personal data about the farmer and their farm businesses was to be collected.

Third, the consumer survey used in the paper from Reference 28 received ethical approval from the University of Newcastle’s Faculty of Science, Agriculture and Engineering Ethics Committee (Ref. No. BH124-1898).

Fourth, when we uploaded the original paper, we stated that no ethical approval was needed. This was an oversight as we had originally obtained this and had overlooked this matter.

Fifth, as suggested by the Reviewer #1, we have very carefully checked for typos in the paper.

Sixth, having addressed the comments of Reviewer #1, we now address the minor comments made by Reviewer #2, namely -

1) Perhaps clarify/rephrase (abstract, lines 35-37): "However, interventions are not economically or societally preferred per se, because their financial and social viability is dependent on the interventions". Do you mean that an evaluation of the financial and social viability of the interventions is needed to recommend/decide what interventions are to be implemented?

We thank Reviewer #2 for this good point and have now amended the text accordingly.

2) Indent lines 89-92

We have done this in the revised text.

3) Suggestion: I would suggest to mention the use of VCA in the introduction section and move the VCA lit review to the material and methods section (making it more succint). I would think this would focus the introduction to the problem in hand whilst touching on how to go about it avoinding dwelling too much on the methods used.

We have now taken this well thought out point on board and, as well as shortening the literature review on VCA, moved most of it to the Materials and Methods section.

4) Description of intervention scenarios: Improved hygiene in pig fattening. Based on results from previous research the amount of meat produced per pig space unit per year is calculated to be 22%. Presumably this is an average/single case study. Do you have any information about the level of certainty of this figure (i.e. how much it could vary depending on circumstances)?

The Reviewer is, indeed, correct as the figure is an average number derived from an experiment. The coefficient of variation for the difference of mass of meat at a given time point was 37%. A sentence indicating this has been added in the text and in the title of Table 1.

5) Table 2 shows change in production parameters resulting from interventions contributing to piglet viability. As above do you have any indication on how senstitive these parameters may be to circumstances? Providing some information on the level of certainty on the parameters used would be helpful for ascertain the defree of confidence of the results obtain. This does not mean that the results obtained here are irrelevant. On the contrary, I think results obtained are very informative for managers and policy makers, but by providing information (when possible) on the level of confidence on the parameters used would give further clarity of the results obtained.

Information about the statistical significance of change reported in Table 2 was provided in cases where it was supplied in the cited source. If this information was unavailable, coefficient of variation for the parameter before the change is provided, where available.

6) Appendix 1 (lines 1300-1303): Perhaps clarify that you are referring a situation with intervention in place

As suggested by the Reviewer, we have now revised these lines to make this clearer.

7) Line 1316: Would it not be P1 instead of P0 (for D0)

The notation has now been corrected; the correct price-quantity pairs should have been P1, Q1 and P0, Q0.

8) Line 1320: "requires that the intervention increases..." Would it not be changes instead of increases? Market prices and quantities can change either with an increase or a decrease of production costs.

We thank the Reviewer for pointing out this typographical error. The sentence has now been revised as suggested.

9) Line 1340 do you mean equilibrium (P0,Q0) instead of origin?

We have now revised the sentence to read as ‘by the area of the triangle on the right side of the vertical line on the origin (0), below …..’ instead of ‘….. left side ….’.

10) Lines 1355 and 1358: Would it be better to put Q1-Q0 instead of change in Q1

The subscripts associated with the change of Q may be confusing in this case. As a result, the relevant lines have been revised as helpfully suggested by the reviewer.

11) Line 367: Just chacking the last term of the change in producer surplus equation. Does it not need to be multiplied by 1/2?

Following this important point made by the reviewer, we have now revised the text so that the multiplier is now after the + sign, in front of the last term. We have also double-checked the logic of the equation.

12) Table 3 shows average figures. As above, would it be possible to say something about the distribution of these parameters? How would results be affected if these parameters change? I am not asking to conduct a sensitivity analysis (although you could do so), but to shed some light by commenting, if possible, on the sensitivity of results and implications of pig interventions to the value of production to variation in the parameter values used.

Table 3 (illustrated in Figure 2) represents values for 2017. Some of these parameters will vary from year to year, or between farms. For most of the parameters we did not have information about between farms variation. We have carried out a sensitivity analysis for the most important parameters, and added sentences briefly explaining insights from this analysis just above Table 3 and just above Table 7.

---

## [Decision Letter · Decision Letter 1]

23 Mar 2020

A value chain analysis of interventions to control production diseases in

the intensive pig production sector

PONE-D-19-30807R1

Dear Prof. Tranter,

We are pleased to inform you that your manuscript has been judged scientifically suitable for publication and will be formally accepted for publication once it complies with all outstanding technical requirements.

With kind regards,

Nikolaos Georgantzis, Dr.

Academic Editor

PLOS ONE

Additional Editor Comments (optional):

Reviewers' comments:

Reviewer's Responses to Questions

**Comments to the Author**

1. If the authors have adequately addressed your comments raised in a previous round of review and you feel that this manuscript is now acceptable for publication, you may indicate that here to bypass the “Comments to the Author” section, enter your conflict of interest statement in the “Confidential to Editor” section, and submit your "Accept" recommendation.

Reviewer #1: All comments have been addressed

Reviewer #2: All comments have been addressed

2. Is the manuscript technically sound, and do the data support the conclusions?

Reviewer #1: Yes

Reviewer #2: Yes

3. Has the statistical analysis been performed appropriately and rigorously? 

Reviewer #1: Yes

Reviewer #2: Yes

4. Have the authors made all data underlying the findings in their manuscript fully available?

Reviewer #1: Yes

Reviewer #2: Yes

5. Is the manuscript presented in an intelligible fashion and written in standard English?

Reviewer #1: Yes

Reviewer #2: Yes

6. Review Comments to the Author

Reviewer #1: I am happy with how the authors addressed my comments. The revised paper is now expected to make an important contribution to the literature.

Reviewer #2: All comments have been addressed satisfactorily, thank you. I believe the manuscript studies an important issue within animal health and welfare using a value chain analysis and pointing out future steps to be taken in future research.

7. PLOS authors have the option to publish the peer review history of their article (what does this mean?). If published, this will include your full peer review and any attached files.

Reviewer #1: No

Reviewer #2: No

---

## [Editor Report · Acceptance letter]

26 Mar 2020

PONE-D-19-30807R1 

A value chain analysis of interventions to control production diseases in
the intensive pig production sector 

Dear Dr. Tranter:

I am pleased to inform you that your manuscript has been deemed suitable for publication in PLOS ONE. Congratulations! Your manuscript is now with our production department. 

With kind regards,

on behalf of

Prof. Nikolaos Georgantzis 

Academic Editor

PLOS ONE